# Analysis of the Formation Mechanism of Medium and Low-Temperature Geothermal Water in Wuhan Based on Hydrochemical Characteristics

Zhibin Yin [1,2], Xuan Li [1,3,*], Changsheng Huang [1,*], Wei Chen [1], Baoquan Hou [4], Xiaozhe Li [1,2], Wenjing Han [1,2], Pingping Hou [1,2], Jihong Han [1,2], Chonghe Ren [1,2], Jin Zou [1,2], Shan Hua [5], Liansan Xu [5] and Ziliang Zhao [6]

1   Wuhan Center of Geological Survey CGS, Wuhan 430205, China
2   Institute of Geological Survey, China University of Geosciences, Wuhan 430074, China
3   School of Environment Studies, China University of Geosciences, Wuhan 430074, China
4   Tianjin Municipal Engineering Design & Research Institute, Tianjin 300392, China
5   The Institute of Hydrogeology and Engineering Geology of Wuhan, Hubei Province Geological Survey, Wuhan 430051, China
6   Fourth Geological Team of Hubei Geological Bureau, Xianning 437100, China
*   Correspondence: xuanli@cug.edu.cn (X.L.); cshuang@cug.edu.cn (C.H.)

**Abstract:** Wuhan and its surrounding areas have obvious geothermal spring outcrops, which are unexplored potential geothermal resources. The degree of geothermal resource development in Wuhan is low, and there is a lack of systematic research on their hydrochemical characteristics and formation mechanism. The Wuhan area is bounded by the Xiang-Guang fault, the South Qinling-Dabie orogenic belt in the north, and the Yangtze landmass in the south, with Silurian and Quaternary outcrops and little bedrock outcrops. The Silurian is the main water barrier in the region, which separates the upper Triassic and Paleogene as shallow aquifers and the lower Cambrian and Ordovician as deep aquifers. Different strata are connected by a series of fault structures, which constitute Wuhan's unique groundwater water-bearing system. Eleven geothermal water (23~52 °C) and six surface water samples (around 22 °C) were collected from the study area. The geothermal water in the study area is weakly alkaline, with a pH of 7.04~8.24. The chemical type of geothermal water is mainly deep $SO_4^{2-}$ with a higher TDS and shallow $HCO_3^-$ type water with a lower TDS. Isotopic analysis indicates that atmospheric precipitation and water-rock interaction are the main ionic sources of geothermal water. The chemical composition of geothermal water is dominated by ion-exchange interactions and the dissolution of carbonates and silicates. The characteristic coefficients, correlation analysis, water chemistry type, recharge elevation, geothermal water age, reservoir temperature, and cycle depth were also analyzed. The performance was similar in the same geothermal reservoir, which could be judged as an obviously deep and shallow geothermal fluid reservoir, and the genetic conceptual model of Wuhan geothermal was preliminarily deduced. DXR-8 and DXR-9 had the best reservoir conditions, hydrodynamic conditions, rapid alternation of water bodies, and large circulation depth, which is a favorable location for geothermal resource development and will bring considerable economic and social benefits.

**Keywords:** geothermal fluids; hydrogeochemistry; formation mechanism

## 1. Introduction

In order to promote green development and implement the goal of carbon peaking and carbon neutrality, more than 80 countries around the world are actively involved in the development and research of green energy [1,2]. Geothermal resources integrate heat, water, and minerals in a highly utilizable, low-cost, and environmentally non-polluting way and can be used for power generation, heating, aquaculture, medical care, and spa bathing,

with considerable economic and environmental benefits [3–6]. Hydrological and hydrogeochemical methods have been used in the early stages of geothermal field exploration, where hydrochemical and isotopic signatures often retain detailed chemical information about the processes of the formation and evolution of the geothermal system, helping to analyze the origins, formation conditions, and geochemical processes of geothermal fluids, as well as to evaluate the recharge sources, ages, classification of hydrochemical types, and analysis of water-rock equilibrium states, reservoir temperatures, and circulation depths [4,7–12].

China is a country rich in geothermal resources, and medium and low-temperature geothermal resources exist widely throughout the country, especially in central China, with great potential for exploitation. It is important to reveal hydrogeochemical processes, analyze geothermal genesis, and explore and develop geothermal mineral resources [13]. Wuhan has carried out several geothermal investigations (surveys), comprehensive geothermal research, and other work. Geothermal spring outcropping conditions and research bases are good, and the available geothermal investigation data indicate that the heat source is the heat production from the decay of radioactive elements in the crustal rock. The deep Cambrian-Ordovician strata in the Xianning, Tianmen and northern Qianjiang, Hanchuan areas have the formation of large karst heat storage conditions [14]. However, there is a lack of systematic research on the study area's hydrochemical characteristics and genesis mechanism, which, to a certain extent, restricts the development and utilization of geothermal resources.

In this paper, the authors collected and gathered surface water and underground hot water samples from the study area and analyzed the samples using the Durov diagram, characteristic coefficient analysis, correlation analysis, isotope characterization, thermal storage temperature, and circulation depth calculation. These analyses were performed to investigate the chemical characteristics and causal mechanisms of geothermal water in Wuhan, to provide some basis for the efficient development and utilization of geothermal resources in the study area.

## 2. Geological and Hydrological Settings of the Study Area

Wuhan is located in Hubei Province in central China, at the confluence of the Yangtze River and Han River, with geographical coordinates of $113°41'~115°05'$ E and $29°58'~31°22'$ N (Figure 1). The city has a land area of 8569.15 km$^2$. Wuhan is located between the Dabie Mountain in northeastern Hubei and the Mufu Mountain in southeastern Hubei, in the eastern part of the Jianghan Plain (the alluvial plain where the Yangtze and Han rivers meet). The middle reaches of the Yangtze River, in the center of a semi-enclosed basin surrounded by mountains on three sides, has low-lying terrain, and the Yangtze and Han rivers intersect there; the general topography is high in the north and low in the south, with undulating terrain between hills and plains, forming a geomorphological pattern dominated by plains, with intermittent and continuous remnants of hills and lakes around. It belongs to the northern edge of the subtropics, with a mild climate, abundant rainfall, and monsoon climate characteristics. The monsoon alternates significantly between winter and summer, with long winters and summers, short springs and autumns, and high temperatures in summer. June has the most precipitation of all months of the year.

The study area spans two primary tectonic units—the Yangzi land mass and the Qinling-Dabie orogenic belt—and is roughly bounded by the Xiangguang Fault along the Yangtze River. The surface is mainly covered by loose Quaternary accumulations, and the Paleozoic strata are strongly deformed, mainly due to the Indo-Yanshan orogenic movement. It mainly reveals the Lower Yangzi-Aurora-Triassic sedimentary strata, Cretaceous-Paleocene, and Quaternary strata. The Yanshan period magmatic activity is frequent, and the Xia Dian rock body represents medium-acidic intrusive bodies. The stratum outcrops in this area are complete—from Cambrian to Quaternary, good outcrops can be found in the Wuhan area. The Cambrian and Ordovician belong to carbonate rocks, and the karst system is developed, which is a mature aquifer system. The Silurian overlies the Ordovician. The Silurian strata in the region, mainly shale and sandy shale, are widely

distributed. Their poor permeability and low thermal conductivity make them a good regional water barrier and heat insulator. Based on the lithology of the Silurian, it can be inferred that the Silurian is a good water-proof stratum, which can be separated from the upper and lower strata in the hydraulic system. It divides Wuhan into two types of underground aquifers, deep and shallow, which are Cambrian-Ordovician strata and two main aquifers of the Devonian-Third system. The lower part is based on the Silurian system, and the upper part is attached to the sedimentary layer of the Tertiary and Quaternary system, which has good water communication with the surface (It is indicated in the conceptual model diagram of geothermal genesis). From the Silurian to the surface of the earth, there are no water-isolated strata with a very wide distribution range. The Permian and Triassic strata above Silurian belong to sedimentary strata with many rock cracks and good water conservancy communication with the surface.

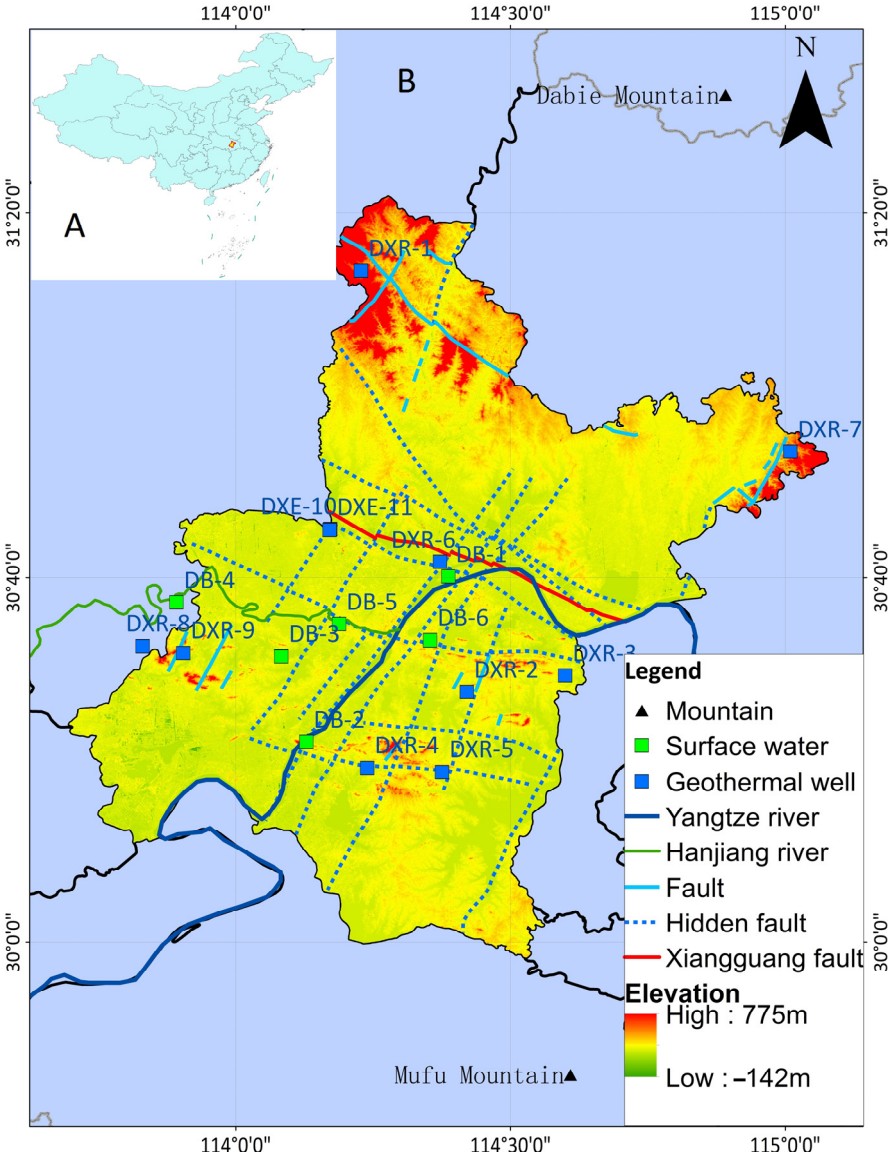

**Figure 1.** Geological sketch and sampling point location map of the study area. Figure **A**: The main purpose is to indicate the specific location of the study area in China. Figure **B**: It contains information on elevation and fault structure and points out the location of sampling points of geothermal wells and surface water in detail.

### 3. Sampling and Analysis

Eleven sets of geothermal and six sets of surface water samples were collected and analyzed in Wuhan and at the border with other cities in 2021; the Geothermal water samples were all from geothermal wells. The sample sampling locations are shown in Figure 1. Before sampling, water was filtered in situ using a 0.45 µm membrane. Water was filtered and stored in sample bottles until $^2$H(D) and $^{18}$O isotopes were analyzed. Water chemistry parameters, such as temperature, pH, EC, and ORP, were tested in situ using a calibrated SMART TROLL multiprobe and conductivity (EC) meter. The electrodes were rinsed with distilled water before measuring each sample's pH to reduce inter-sample contamination.

DB-1 and DB-2 are derived from Yangtze River water; DB-3 from Zhiyin Lake water; DB-4 and DB-5 from the Han River water; and DB-6 is derived from East Lake water. DXR1–DXR11 are all derived from geothermal water obtained from geothermal drilling. All sampling sites were taken for full analysis, and trace elements, hydrogen and oxygen isotope water samples, and carbon 14 age isotope water samples were collected from 11 geothermal wells. For 14 water sample sites, each site was taken 2 L of water: half was acidified and preserved with 5–10 mL 1:1 nitric acid; Half sealed and stored without handling. All the water samples were sent to the testing institution within 10 days, and the test analysis was completed. Hubei Geological and Environmental Laboratory did a full analysis and testing of trace elements. The testing of hydrogen and oxygen isotopes included two parts: $^2$H(D) and $^{18}$O analysis at 33 sampling points. The experimental instrument for D and $^{18}$O testing used a water isotope analyzer (LGR, USA, IWA-45EP), which was completed by the Geological Survey Experiment Center of the China University of Geosciences (Wuhan). The 11 sets of carbon 14 age isotope samples were tested by the Groundwater Mineral Water and Environmental Testing Center of the Institute of Hydrogeology and Environmental Geology, Chinese Academy of Geological Sciences, using an ultra-low background liquid scintillation spectrometer 1220 from PE. The test results were calculated after isotope fractionation correction, and the data quality was reliable. Table A1 in Appendix A shows the water temperature at the sampling point, the elevation of the sampling point, and some test results.

### 4. Results and Discussion

#### 4.1. Hydrochemistry Characteristics

The geothermal water in the study area is weakly alkaline, with pH 7.04~8.24, and the chemical type of geothermal water in the study area is relatively simple, as can be seen from the Durov diagram [13] (Figure 2). DXR-8 and DXR-9 are of the $SO_4$-Ca·Mg type. DXR-10 and DXR-11 are $SO_4$-Ca·Na and had higher TDS in four places—2845.07 mg/L and 2867.57 mg/L for DXR-8 and DXR-9, respectively, and 1553.50 mg/L and 1552.40 mg/L for DXR-10 and DXR-11, respectively. DXR-7 is $HCO_3$·$SO_4$-Na type water. The anions of the rest of the hot water sites are mainly $HCO_3^-$, and TDS is not very different: 178.70~659.40 mg/L, among which DXR-6 and DXR-7 are relatively high, at 643.15 and 659.40 mg/L, respectively. Surface water anions are mainly $HCO_3^-$, and TDS is low.

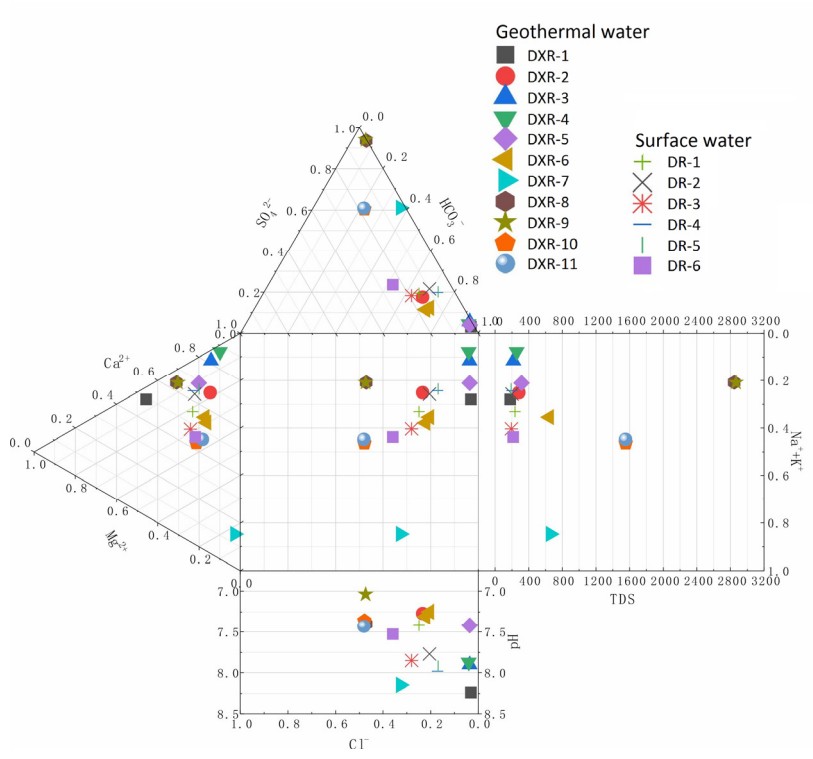

**Figure 2.** Durov plot of water sample in the study area.

### 4.2. Characteristic Coefficient Analyses

The material concentration ratios of certain ions in groundwater can be used to analyze the state of groundwater and related geological effects, so the transport and accumulation of geothermal fluids can be analyzed by some specific ion concentration ratios. The commonly used characteristic coefficients are denaturation coefficient $\gamma(Na)/\gamma(Cl)$, calcium chloride coefficient $\gamma(Cl)/\gamma(Ca)$, desulfurization coefficient $\gamma(Na)/\gamma(K)$, etc. The characteristic coefficients of geothermal water in the study area and those of the surface water used as a comparison are calculated in Table 1.

**Table 1.** Calculation results of geothermal water sample characteristic coefficients.

| ID | $\gamma(Na)/\gamma(Cl)$ | $\gamma(Cl)/\gamma(Ca)$ | $100 \times \gamma(SO_4)/\gamma(Cl)$ |
|---|---|---|---|
| DXR-1 | 3.371 | 0.056 | 107.810 |
| DXR-2 | 1.111 | 0.448 | 59.354 |
| DXR-3 | 5.002 | 0.019 | 370.601 |
| DXR-4 | 1.423 | 0.040 | 130.075 |
| DXR-5 | 7.128 | 0.043 | 144.767 |
| DXR-6 | 1.656 | 0.512 | 42.864 |
| DXR-7 | 38.743 | 0.304 | 1449.741 |
| DXR-8 | 16.362 | 0.009 | 17176.867 |
| DXR-9 | 12.206 | 0.013 | 11672.348 |
| DXR-10 | 2.064 | 0.773 | 171.223 |
| DXR-11 | 2.052 | 0.727 | 172.644 |
| DB-1 | 1.398 | 0.509 | 63.415 |
| DB-2 | 1.410 | 0.296 | 110.491 |
| DB-3 | 1.330 | 0.749 | 48.165 |
| DB-4 | 1.648 | 0.219 | 135.937 |
| DB-5 | 1.786 | 0.205 | 146.979 |
| DB-6 | 1.181 | 1.061 | 48.921 |

The ratio of Na/Cl is an important index that can reflect the formation's confinement, the degree of groundwater concentration and metamorphism, activity, and the hydrogeochemical environment of the reservoir. Generally, the smaller the value of $\gamma Na^+/\gamma Cl^-$, the better the confinement of the formation water and the more concentrated and deeper the metamorphism. Conversely, the larger the value, the stronger the infiltration water activity and the more the formation water is influenced by the infiltration water. If the groundwater is mainly dissolved and filtered from rock salt-bearing strata, the sodium-chloride coefficient should be close to 1 [15,16]. The denaturation coefficient of geothermal water sites in the study area ranges from 1.111 to 38.743. The denaturation coefficient of surface water sites compared with geothermal water sites ranges from 1.181 to 1.786. The large denaturation coefficient of geothermal water indicates that the geothermal reservoir of geothermal water is not highly closed and is closely connected with external atmospheric precipitation. Among them, DXR-2, DXR-4, and DXR-6 are 1.111, 1.423, and 1.656, respectively, with relatively good confinement; DXR-8 and DXR-9 are 16.362 and 12.206, respectively, with relatively open reservoirs; and the DXR-7 geothermal reservoir is the most open with a metamorphic coefficient of 38.743.

The calcium chloride coefficient ($\gamma Cl^-/\gamma Ca^{2+}$) characterizes the hydrodynamic characteristics of groundwater, and the larger its value, the poorer the groundwater flow conditions and the slower the groundwater flow. Usually, $Ca^{2+}$ is the characteristic cation in low-TDS water, while $Cl^-$ accumulates in geothermal reservoirs with weak hydrodynamic conditions, so the magnitude of ($\gamma Cl^-/\gamma Ca^{2+}$) value can indicate the strength of the hydrodynamic drive of geothermal reservoirs. The calcium chloride coefficients of surface water in the study area ranged from 0.205 to 1.061, and those of geothermal water ranged from 0.009 to 0.773. Overall, the geothermal water hydrodynamic conditions were relatively good. Among them, the characteristic coefficients of DXR-8 and DXR-9 were 0.009 and 0.013, respectively, with the best hydrodynamic conditions, indicating that the degree of water body alternation was faster and the water-rock dissolution filtration was more fully carried out. The hydrodynamic conditions of DXR-1, DXR-3, DXR-4, and DXR-5 were relatively good. The calcium chloride coefficients of DXR-10 and DXR-11 were 0.773 and 0.727, respectively, and the dynamic conditions were relatively poor.

The desulfurization ratio ($100*\gamma(SO_4^{2-})/\gamma(Cl^-)$) can reflect the redox environment of the formation water, the smaller the desulfurization factor, the better the confinement of the formation water. Generally, the desulfurization factor is equal to 1 as the limit value [17]. If the desulfurization ratio is less than 1, it indicates that the formation water is completely reduced and buried in a well-confined area; if the desulfurization factor is greater than 1, it indicates that the reduction is not complete, and the formation water may be affected by shallow oxidation. The geothermal water desulfurization coefficients in the study area ranged from 42.864 to 17,176.867, all of which were much greater than 1. The confinement conditions were poor and influenced by shallow surface oxidation.

### 4.3. Ionic Component Correlation Analysis

Correlation analysis of ions can analyze and infer the hydrogeochemistry of macronutrients in geothermal water, which was analyzed with the correlation analysis module of SPSS 26 (Statistical Product and Service Solutions) software, and the Pearson correlation coefficient was selected [18–20].

Due to the scattered distribution of sampling points, the analysis of water chemistry type shows that the study area is mainly divided into two types of water, one type of anion is mainly $SO_4^{2-}$ and TDS is larger, and the other type of anion is mainly $HCO_3^-$ and TDS is smaller. It indicates that the two types of water have different components of the soluble filtered envelope during the long-term evolution, have different sources, or different water-rock interactions have occurred, so the correlation analysis of the two types of water sample points was conducted separately. (See Tables 2 and 3).

**Table 2.** Geothermal water sample ion correlation analysis (A).

|  | TDS | K$^+$ | Na$^+$ | Ca$^{2+}$ | Mg$^{2+}$ | HCO$_3$$^-$ | SO$_4$$^{2-}$ | Cl$^-$ |
|---|---|---|---|---|---|---|---|---|
| TDS | 1 |  |  |  |  |  |  |  |
| K$^+$ | −0.694 | 1 |  |  |  |  |  |  |
| Na$^+$ | −0.999 | 0.682 | 1 |  |  |  |  |  |
| Ca$^{2+}$ | 0.999 | −0.715 | −0.999 | 1 |  |  |  |  |
| Mg$^{2+}$ | 0.996 | −0.628 | −0.997 | 0.993 | 1 |  |  |  |
| HCO$_3$$^-$ | −0.999 | 0.718 | 0.998 | −0.999 | −0.992 | 1 |  |  |
| SO$_4$$^{2-}$ | 1 | −0.7 | 0.998 | 0.999 | 0.995 | −1 | 1 |  |
| Cl$^-$ | −1 | 0.683 | 1 | −0.999 | −0.997 | 0.999 | −1 | 1 |

**Table 3.** Geothermal water sample ion correlation analysis (B).

|  | TDS | K$^+$ | Na$^+$ | Ca$^{2+}$ | Mg$^{2+}$ | HCO$_3$$^-$ | SO$_4$$^{2-}$ | Cl$^-$ |
|---|---|---|---|---|---|---|---|---|
| TDS | 1 |  |  |  |  |  |  |  |
| K$^+$ | 0.651 | 1 |  |  |  |  |  |  |
| Na$^+$ | 0.401 | 0.574 | 1 |  |  |  |  |  |
| Ca$^{2+}$ | 0.979 | 0.597 | 0.236 | 1 |  |  |  |  |
| Mg$^{2+}$ | 0.978 | 0.573 | 0.228 | 0.985 | 1 |  |  |  |
| HCO$_3$$^-$ | −0.008 | 0.516 | 0.345 | 0.003 | −0.099 | 1 |  |  |
| SO$_4$$^{2-}$ | 0.985 | 0.53 | 0.302 | 0.97 | 0.982 | −0.165 | 1 |  |
| Cl$^-$ | 0.267 | 0.653 | 0.763 | 0.178 | 0.158 | 0.471 | 0.13 | 1 |

Probably due to the small number of water sample points in Class A, the reference value of the analytical results is poor, and all the ions show a good correlation with each other. According to the above water sample test results, the SO$_4$$^{2-}$ content of Class A water is very high, much higher than other ions. It is presumed that the geothermal water flowed through the paste rock layer for full dissolution and filtration during the ascending process, which greatly increased the SO$_4$$^{2-}$ content. The lower section is the main production layer. From top to bottom, it can be divided into large paste layers: I oil group, II oil group, and mud septum and III oil group [21], which verifies the above view.

The analysis results of water samples in group B showed obvious patterns in which TDS was strongly correlated with Ca$^{2+}$, Mg$^{2+}$, and SO$_4$$^{2-}$, indicating that the main contributing ions of geothermal water TDS in the study area were Ca$^{2+}$, Mg$^{2+}$, and SO$_4$$^{2-}$. In contrast, the correlation coefficient between TDS and HCO$_3$$^-$ was −0.008, a very poor correlation, and none of the correlations between HCO$_3$$^-$ and the other ions were strong, indicating that HCO$_3$$^-$ does not have the same origin as the other ions. It is presumed that the content of HCO$_3$$^-$ ions are greatly increased due to the mixing of surface water or shallow cold water, but the existence of the same material source of HCO$_3$$^-$ as Ca$^{2+}$ and Mg$^{2+}$ cannot be excluded after removing the effect of cold water mixing on HCO$_3$$^-$ content carbonates, which are rich in dolomite and limestone. It is speculated that the dissolution of dolomite and calcite may have occurred (Formulas (1) and (2)). Still, the cold water mixing interfered with the correlation presentation of ions.

$$\text{Dolomite stone } CaMg(CO_3)_2 + 2CO_2 + 2H_2O \rightarrow Ca^{2+} + Mg^{2+} + 4HCO_3^- \tag{1}$$

$$\text{Calcite } CaCO_3 + CO_2 + H_2O \rightarrow Ca^{2+} + 2HCO_3^- \tag{2}$$

$$\text{Gypsum } CaSO_4 + 2H_2O \rightarrow Ca^{2+} + SO_4^{2-} \tag{3}$$

The correlation coefficient between Na$^+$ and Cl$^-$ was 0.763, while the correlation coefficients between Ca$^{2+}$ and Mg$^{2+}$ and Cl$^-$ were only 0.178 and 0.158, indicating that Na$^+$ and Cl$^-$ have the same ionic origin and may originate from the dissolution of rock salt, while Ca$^{2+}$ and Mg$^{2+}$ have different ionic origin from Cl$^-$. The correlation coefficient of SO$_4$$^{2-}$ with Ca$^{2+}$ is 0.97, presumably from the dissolution of gypsum (Formula (3)), and the

correlation coefficient of $SO_4^{2-}$ with $Mg^{2+}$ is 0.982 since sodium, potassium, magnesium, and calcium salts are all extremely water-soluble salts.

### 4.4. Isotopes Characteristics

#### 4.4.1. Source Recharge

Different bodies of water in the water cycle have their characteristic isotopic compositions due to different genesis, and they are enriched in different levels of the heavy isotopes hydrogen (D) and oxygen ($^{18}$O). By analyzing the isotopic "traces" of water bodies in different environments, it is possible to trace their formation and transport patterns [22,23].

One can infer the geothermal water source based on the study area's $\delta^{18}$O and $\delta$D data from water sampling sites. We selected the Chinese atmospheric precipitation line ($\delta$D = 7.9$\delta^{18}$O + 8.2) and the Wuhan City area atmospheric precipitation line ($\delta$D = 8.29$\delta^{18}$O + 7.44) [24] for analysis. We mapped the deuterium-oxygen isotope distribution of surface and subsurface hot water in Wuhan City (Figure 3).

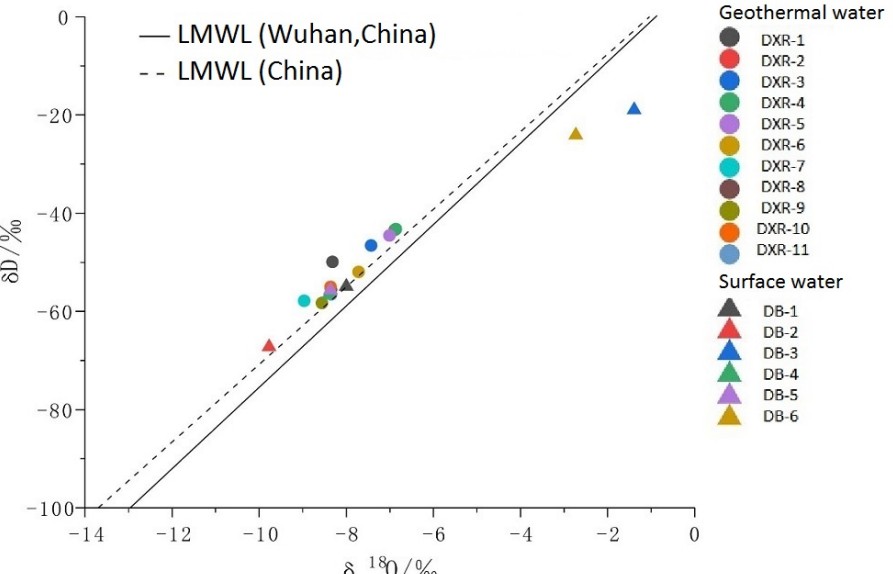

**Figure 3.** Distribution map of deuterium oxygen isotopes in surface and underground hot water.

The results show that the underground hot water samples and some surface water sample points are distributed near the atmospheric precipitation process line, indicating that the geothermal water in the study area comes from the infiltration recharge of atmospheric precipitation. Two surface water sample sites were offset and showed anomalously high values of oxygen isotopes: $\delta^{18}$O/‰ reached −2.73 and −1.39 for DB-6 and DB-3, respectively, and the reasons for the anomalies were not clear. Since it has no influence on the conclusion of the article, it will not be discussed here.

#### 4.4.2. Recharge Elevation

The $\delta^2$H and $\delta^{18}$O values of atmospheric precipitation decrease with increasing altitude, with an elevation effect. As the altitude varies, the temperature will also vary, leading to changes in isotopic fractionation coefficients. This effect varies considerably around the world depending on climatic and topographic conditions. This elevation effect makes it possible to infer the location and height of groundwater recharge areas [11,23].

Using the elevation effect of $\delta^2$H and $\delta^{18}$O values of atmospheric precipitation, the elevation of the recharge zone of underground hot water can be estimated with the following Formula (4).

$$H = (\delta R - \delta P)/K + h \qquad (4)$$

In the equation, H is the elevation of the recharge area (m); h is the elevation of the sampling point (m); $\delta$R is the $\delta$D or $\delta^{18}$O value of the water sample at the sampling point; $\delta$P is the $\delta$D or $\delta^{18}$O value of the atmospheric precipitation (recharge water) near the reference point; and K is the elevation gradient of the $\delta$D or $\delta^{18}$O of the atmospheric precipitation ($\delta$/100 m).

By studying the variation characteristics of $\delta$D and $\delta^{18}$O of atmospheric precipitation in Wuhan, Gu [25] determined that the variation range of $\delta$D for atmospheric precipitation in Wuhan was $-86.8\permil$ to $+4.2\permil$, the variation range of $\delta^{18}$O was $-12.18\permil$ to $+0.12\permil$, and the annual average values of both were $-41.2\permil$ and $-5.82\permil$, respectively. Since they have not studied the elevation effect of rainfall in the Wuhan area, we can only roughly deduce the recharge elevation of subsurface hot water in Wuhan city with the help of the analogy method. Therefore, $\rho$ is taken here from the research results of Wang Yingzhao [26] on the spatiotemporal evolution of stable isotopes of atmospheric precipitation in western Hubei. The average decreasing rate of $\delta^{18}$O for atmospheric precipitation with elevation in western Hubei is taken as $-0.17 \pm 0.05\permil/100$ m, $\delta$D is taken as $-1.20 \pm 0.35\permil/100$ m, and the formulas for estimating the elevation of recharge area using $\delta^{18}$O and $\delta$D values are obtained respectively.

$$H = \frac{\delta^{18}O - (-5.82)}{-0.17} \times 100 + h \tag{5}$$

$$H = \frac{\delta D - (-41.2)}{-1.2} \times 100 + h \tag{6}$$

The $\delta^{18}$O and $\delta^2$H values of the underground hot water in the study area were substituted into Formulas (5) and (6), respectively. The estimated results are shown in Table 4. The elevation range of the underground hot water recharge area obtained from Equation (5) is 631.08–1915.33 m, and the recharge elevation range obtained from equation 6 is 188.49–1449.89 m.

**Table 4.** Geothermal water recharge zone elevation calculation.

| Id | Sampling Elevation | $\delta$D/‰ | $\delta^{18}$O/‰ | Formula (5) | Formula (6) | Formula (7) |
|---|---|---|---|---|---|---|
| DXR-1 | 108.00 | −49.92 | −8.32 | 1578.59 | 834.67 | 1146.00 |
| DXR-2 | 20.88 | −43.42 | −6.89 | 650.29 | 205.88 | 821.00 |
| DXR-3 | 10.34 | −46.56 | −7.43 | 957.40 | 457.01 | 978.00 |
| DXR-4 | 19.32 | −43.23 | −6.86 | 631.08 | 188.49 | 811.50 |
| DXR-5 | 45.00 | −44.53 | −7.01 | 745.00 | 322.50 | 876.50 |
| DXR-6 | 23.00 | −51.95 | −7.72 | 1140.65 | 918.83 | 1247.50 |
| DXR-7 | 62.39 | −57.85 | −8.97 | 1915.33 | 1449.89 | 1542.50 |
| DXR-8 | 21.00 | −55.59 | −8.35 | 1509.24 | 1220.17 | 1429.50 |
| DXR-9 | 21.64 | −58.28 | −8.56 | 1633.40 | 1444.97 | 1564.00 |
| DXR-10 | 7.54 | −55.02 | −8.36 | 1501.66 | 1159.21 | 1401.00 |
| DXR-11 | 5.38 | −56.51 | −8.35 | 1493.62 | 1281.21 | 1475.50 |

In addition, the elevation effect of atmospheric precipitation $\delta^2$H in China is also shown in Formula (7).

$$\delta D = -0.02H - 27 \tag{7}$$

In the equation, H is the elevation of the recharge area (m). The calculated results are listed in Table 4, and the obtained elevation range of the geothermal water recharge area is 811.50 m–1564.00 m.

Wuhan is located in the center of a semi-enclosed basin surrounded by mountains on three sides, with Dahong Mountain at 1055 m, Dabie Mountain at 1777 m, and Mufu Mountain at 1595.6 m. Based on the calculation results of Equations (5)–(7), combined

with the topography of Wuhan, it is assumed that geothermal water comes from the three surrounding peaks for nearby recharge.

### 4.4.3. Age of Isotopes $^{14}$C

The results of the $^{14}$C dating [27] of water samples are shown in Table 5. The ages of water samples in the study area vary widely, with DXR-1, DXR-2, DXR-3, DXR-4, and DXR-5 geothermal water having relatively small ages and relatively fast circulation rates, DXR-6, DXR-7, DXR-8, DXR-9, DXR-10, and DXR-11 having relatively large apparent ages, and DXR-7 having the largest age that reached 41.76 ka. The groundwater experienced a long period of circulation.

**Table 5.** Apparent age of geothermal water.

| Id | Age ka |
| --- | --- |
| DXR-1 | $4.98 \pm 0.22$ |
| DXR-2 | 2.31 |
| DXR-3 | 8.04 |
| DXR-4 | 5.9 |
| DXR-5 | $8.92 \pm 0.23$ |
| DXR-6 | $17.12 \pm 1.15$ |
| DXR-7 | 41.76 |
| DXR-8 | 38.03 |
| DXR-9 | 23.5 |
| DXR-10 | 37.71 |
| DXR-11 | 37.99 |

### 4.5. Estimation of Geothermal Reservoir Temperature

#### 4.5.1. Water-Rock Equilibrium State Analysis

The use of the geothermometer method for calculating the temperature of a geothermal reservoir presupposes that a substance used as a geothermal geothermometer and the minerals in the geothermal reservoir are in equilibrium. Therefore, it is necessary to study the equilibrium state of geothermal water and minerals to test the reliability of the geothermometer [28].

Na-K-Mg triangle diagrams are often used to study the degree of water-rock interaction in geothermal systems, to evaluate the state of water-rock equilibrium, and to distinguish between different types of water samples [29]. The geothermal water samples in the study area belong to "immature water" and "partially balanced water" (Figure 4), indicating that the ionic equilibrium between water and rock in the geothermal water samples has not yet been reached. Dissolution is still in progress, or the hot water is diluted by mixing shallow cold water in the process of rising from the deep to the surface, thus lowering the elemental content of the hot water. In this case, a reasonable equilibrium temperature value cannot be calculated using the cationic geothermometer, and a certain degree of deviation will occur. Therefore, the geothermal water samples in the study area are not suitable for calculating the geothermal reservoir temperature using a cationic geothermometer.

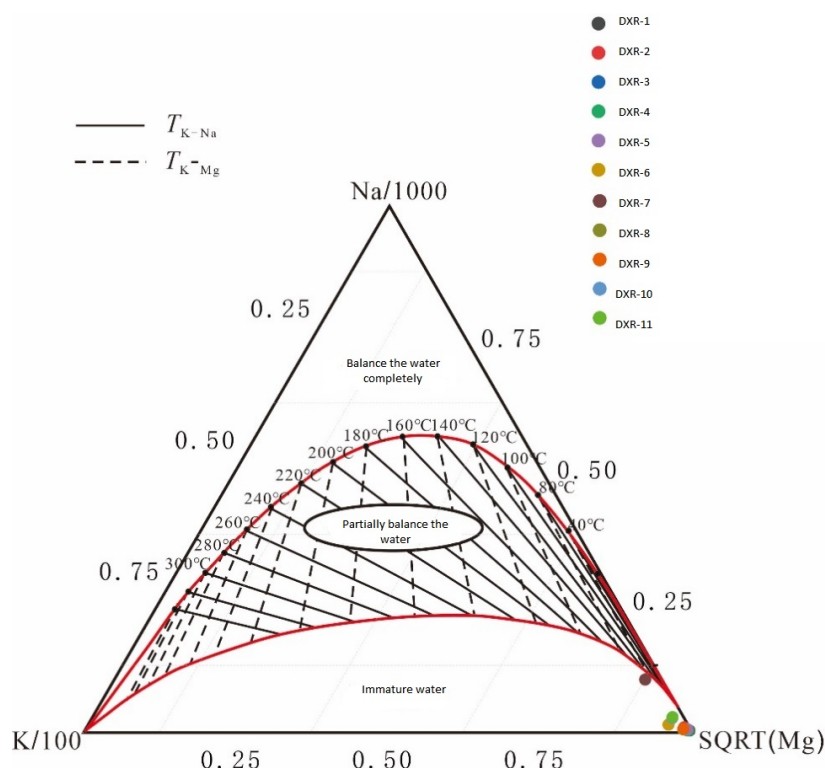

**Figure 4.** Na-K-Mg triangle diagrams of the geothermal fluid in the study area.

4.5.2. Geothermal Reservoir Temperature Estimation

1. Multiple Mineral Equilibrium Approaches

Reed and Spycher proposed using multi-mineral equilibrium diagramming to determine the chemical equilibrium between hydrothermal fluids and minerals in geothermal systems [30,31].

The principle is to dissolve various minerals in water as a function of temperature. If a group of minerals at a particular temperature simultaneously is close to equilibrium, observers can judge the hot water and the group of minerals to have reached equilibrium. The equilibrium temperature is the temperature of the geothermal reservoir temperature.

This paper analyzed 11 geothermal water samples, and the saturation indices of different minerals in the geothermal system at various temperature conditions were calculated using phreeqc [32]. Based on the comprehensive analysis of the lithology and geothermal water chemical characteristics of the study area, minerals such as Anhydrite, chalcedony, chrysotile, quartz, and talc were selected, their mineral saturation indices at 20 °C, 40 °C, 60 °C, 80 °C, 100 °C, 120 °C, 140 °C, 160 °C, and 180 °C were calculated, and T-lg (Q/K) diagrams were plotted (Q is the activity product, K is the equilibrium constant) in Figure 5. We can see that none of the five minerals is simultaneously close to equilibrium at a particular temperature, and most graphic methods do not work well in practical applications. The method can only be used as a basis for a qualitative judgment of geothermal fluid-mineral equilibrium and cannot be used to estimate the geothermal reservoir temperature accordingly.

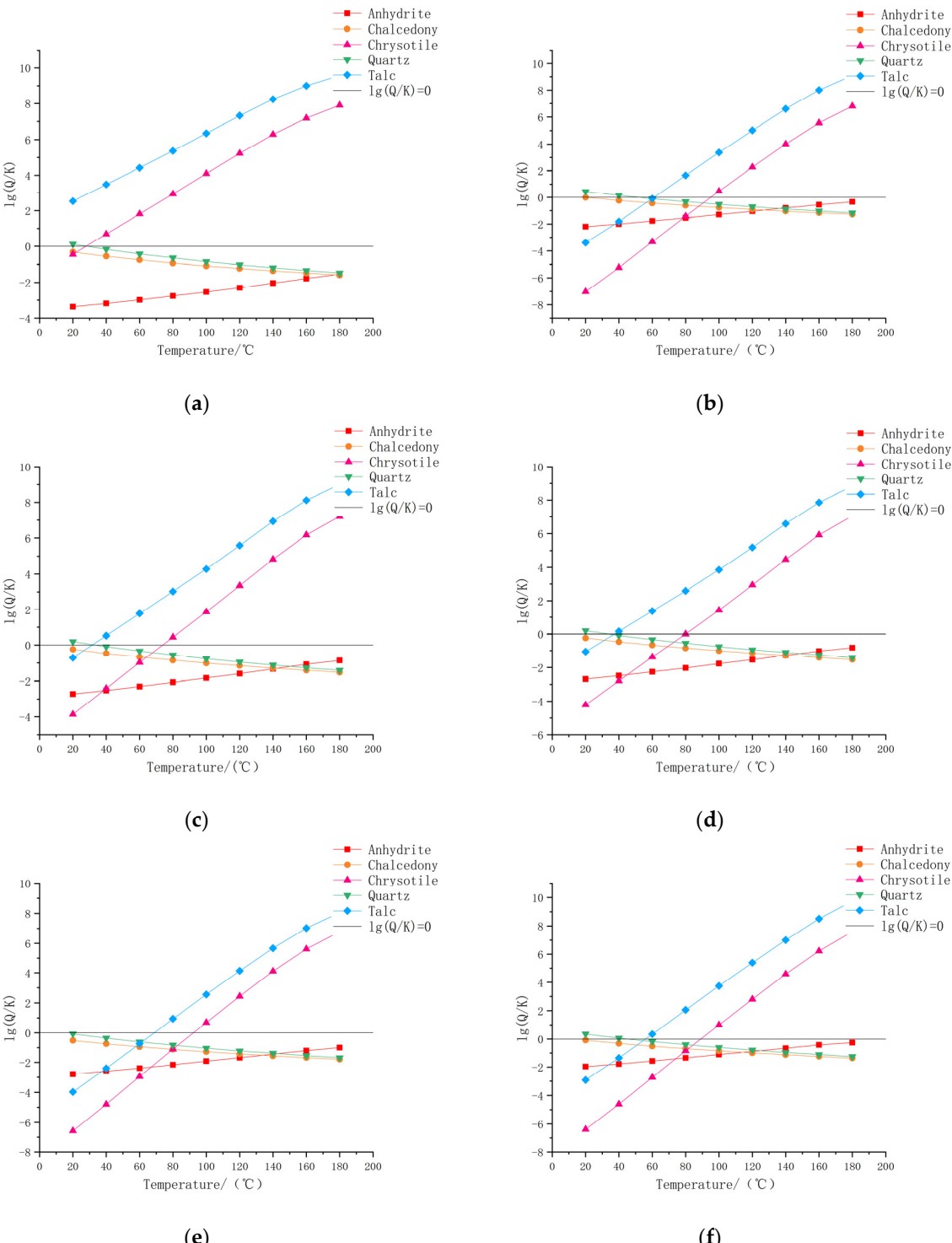

**Figure 5.** *Cont.*

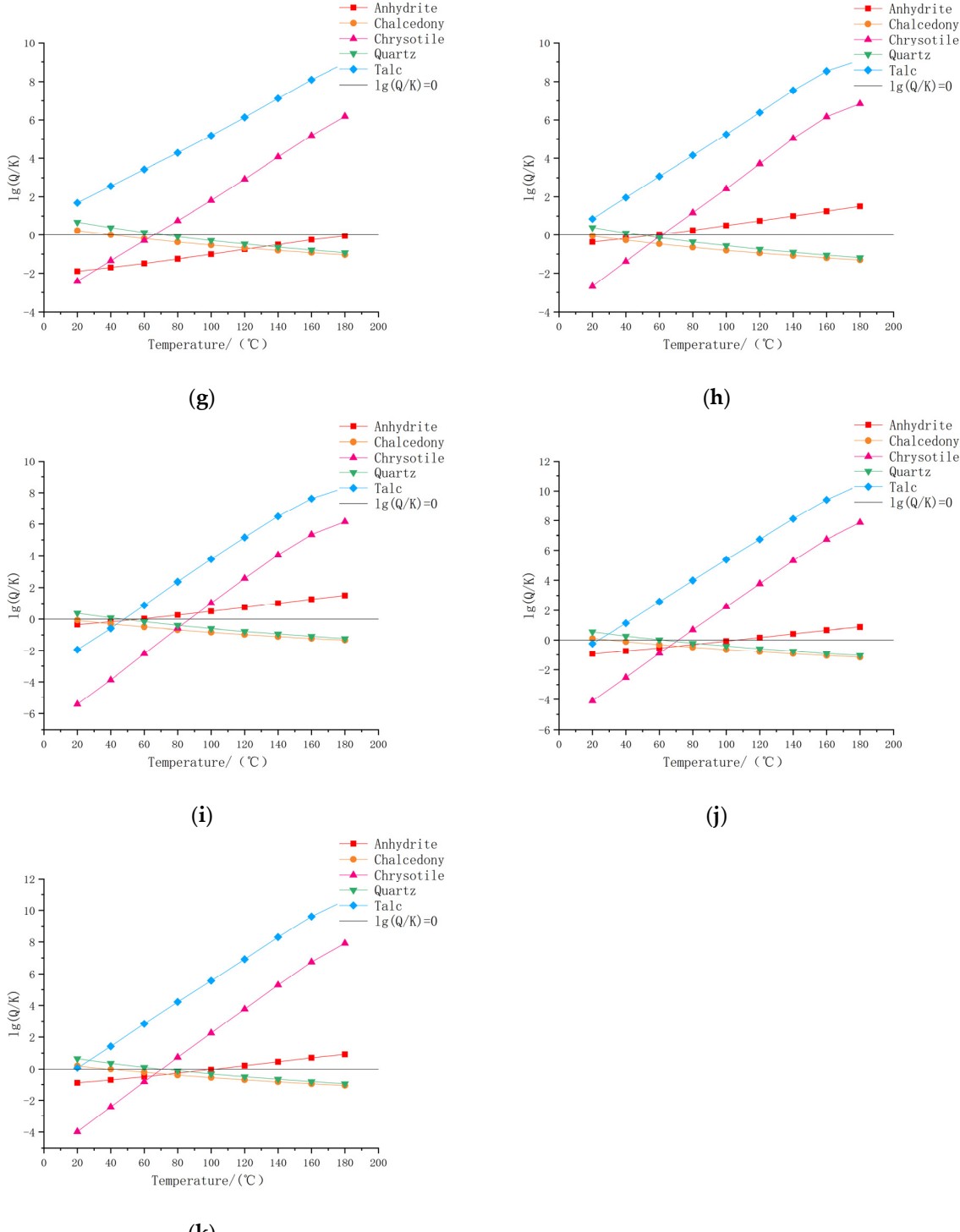

**Figure 5.** Mineral saturation diagram: (**a**) DXR-1; (**b**) DXR-2; (**c**) DXR-3; (**d**) DXR-4; (**e**) DXR-5; (**f**) DXR-6; (**g**) DXR-7; (**h**) DXR-8; (**i**) DXR-9; (**j**) DXR-10; (**k**) DXR-11.

2.    SiO$_2$ Geothermometer Method

The silicon dioxide geothermometer is based on the dissolution-equilibrium theory of silicon minerals in hydrothermal fluids. It calculates the geothermal reservoir temperature reflected by the equilibrium of water-rock interaction in the thermal storage based on the concentration of SiO$_2$ in the geothermal fluid and the solubility in different SiO$_2$ forms. According to relevant research, the applicable range of the SiO$_2$ geothermometer is 20~250 °C. The commonly used SiO$_2$ geothermometers are as follows.

(1) Quartz thermometers (non-vapor loss):

$$T = \frac{1309}{5.19 - \lg(SiO_2)} - 273.15 (Fournier, 1977) \tag{8}$$

(2) Quartz thermometers (maximum vapor loss):

$$T = \frac{1552}{5.75 - \lg(SiO_2)} - 273.15 \tag{9}$$

(3) chalcedony thermometers:

$$T = \frac{1032}{4.69 - \lg(SiO_2)} - 273.15 \tag{10}$$

(4) $\alpha$-cristobalite thermometers:

$$T = \frac{1000}{4.78 - \lg(SiO_2)} - 273.15 \tag{11}$$

(5) $\beta$-cristobalite thermometers:

$$T = \frac{1000}{4.51 - \lg(SiO_2)} - 273.15 \tag{12}$$

(6) Amorphous $SiO_2$ thermometers:

$$T = \frac{731}{4.52 - \lg(SiO_2)} - 273.15 \tag{13}$$

The calculation results show that the calculated values of the amorphous $SiO_2$ geothermometer are all negative, which is not consistent with reality and will not be considered (Table 6). The calculation results of chalcedony, $\alpha$-Cristobalite, and $\beta$-Cristobalite geothermometers have several sets of data lower than the sampling water temperature, indicating that these geothermometers do not apply to this region, and the calculation results are not reliable. The calculated results are 38.04–106.67 °C for Quartz (non-vapor loss) and 52.46–114.24 °C for Quartz (maximum vapor loss), which are close and consistent with the actual situation. The previous paper used phreeqc [32] software to calculate the mineral saturation index (SI) values in geothermal water in the study area. The results showed that the mineral saturation index of quartz ranged from 1.65 to 2.56, all of which were greater than 0, indicating that the quartz minerals in the geothermal water were supersaturated. The $SiO_2$ geothermometer could be used to calculate the geothermal reservoir temperature in this region.

**Table 6.** The $SiO_2$ geothermometer calculates the heat storage temperature.

| Id | Temperature °C | Quartz (Non-Vapor Loss) | Quartz (Maximum Vapor Loss) | Chalcedony | $\alpha$-Cristobalite | $\beta$-Cristobalite | Amorphous $SiO_2$ |
|---|---|---|---|---|---|---|---|
| DXR-1 | 26.00 | 53.76 | 66.89 | 21.36 | 5.08 | 27.68 | −53.91 |
| DXR-2 | 23.00 | 81.32 | 91.79 | 50.08 | 31.47 | 58.77 | −31.32 |
| DXR-3 | 23.10 | 59.85 | 72.44 | 27.65 | 10.87 | 34.46 | −48.98 |
| DXR-4 | 25.60 | 59.85 | 72.44 | 27.65 | 10.87 | 34.46 | −48.98 |
| DXR-5 | 26.00 | 38.04 | 52.46 | 5.29 | −9.74 | 10.43 | −66.44 |
| DXR-6 | 22.60 | 73.45 | 84.73 | 41.80 | 23.88 | 49.77 | −37.85 |
| DXR-7 | 32.00 | 106.67 | 114.24 | 77.12 | 56.19 | 88.34 | −9.85 |
| DXR-8 | 51.30 | 76.79 | 87.72 | 45.30 | 27.09 | 53.58 | −35.09 |
| DXR-9 | 51.00 | 73.45 | 84.73 | 41.80 | 23.88 | 49.77 | −37.85 |
| DXR-10 | 31.30 | 92.30 | 101.56 | 61.71 | 42.12 | 71.45 | −22.11 |
| DXR-11 | 22.90 | 101.92 | 110.06 | 72.00 | 51.53 | 82.72 | −13.93 |

3.  Silicon-Enthalpy Mixing Model

As mentioned before, all the geothermal water samples in the study area fall in "immature water" in the lower right corner of the Na-K-Mg triangle diagram, which may be because the underground hot water has not fully reacted with the surrounding rock or has mixed with the underground cold water in the shallow part. The occurrence of hot and cold water mixing will lead to a decrease in hot water temperature and a change in chemical composition [32].

Based on the enthalpy values of subsurface hot water and surface water and the $SiO_2$ content in water, we can apply the following two equations to find the cold-water mixing ratio.

$$S_cX_1 + S_h(1 - X_1) = S_s \tag{14}$$

$$SiO_{2c}X_2 + SiO_{2h}(1 - X_2) = SiO_{2s} \tag{15}$$

In the equation, X is the cold-water mixing ratio; $S_C$ is the enthalpy of cold water, 22.6 °C (Average value of surface water samples); $S_h$ is the enthalpy of deep hot water; $S_S$ is the enthalpy of geothermal well water; $SiO_{2C}$ is the content of $SiO_2$ of cold water; $SiO_{2h}$ is the content of $SiO_2$ of deep hot water; $SiO_{2S}$ is the content of $SiO_2$ of geothermal well water.

According to the two equations to solve the values of $X_1$ and $X_2$ at different temperatures and plot them versus temperature can be obtained from two curves, the vertical coordinate of the intersection point is the estimated cold water mixing ratio, and the horizontal coordinate is the geothermal reservoir temperature.

After calculation and graphical plotting, except for DXR-8 and DXR-9 (Figure 6), the silicon-enthalpy mixing graphical solutions of other points do not show intersection points, indicating that no cold-water mixing occurs during the rise of underground hot water from depth to the surface at these points. The cold-water mixing ratio of DXR-8 is 71%, and the estimated geothermal reservoir temperature is 121.9 °C; the cold-water mixing ratio of DXR-9 is 68%, and the estimated geothermal reservoir temperature is 114.2 °C, which is higher than the $SiO_2$ geothermometer calculation result. The geothermal reservoir temperatures of DXR-8 and DXR-9 are 121.9 °C and 114.2 °C, respectively, considering the cold water mixing factor correction.

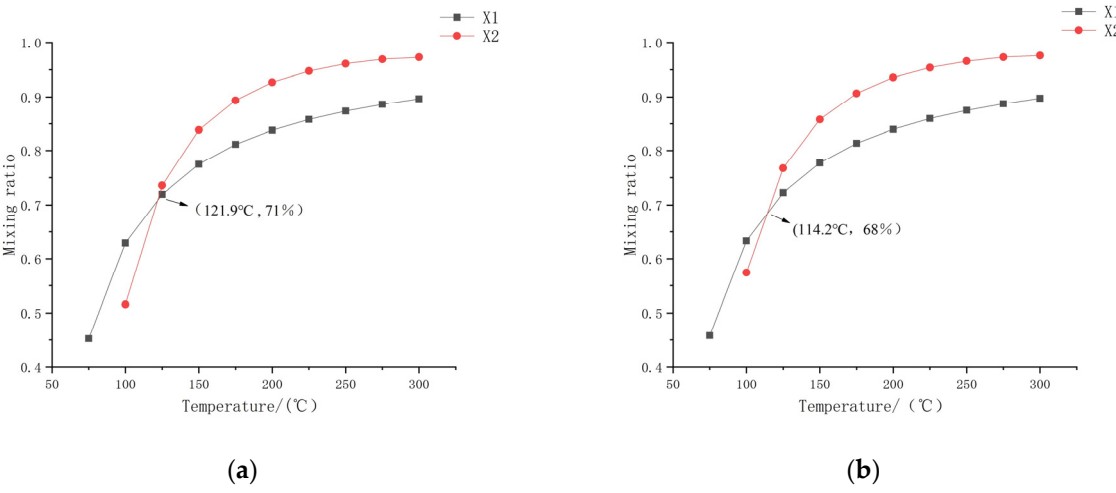

(**a**)            (**b**)

**Figure 6.** Graphical representation of the ratio of mixed cold water and deep hot water in the study area: (**a**) DXR-8; (**b**) DXR-9.

*4.6. Cycle Depth Estimation*

The analysis shows that the infiltration of atmospheric precipitation forms the geothermal water in the study area after deep circulation to obtain deep heat flow heating so that the geothermal reservoir temperature can estimate the depth of geothermal water circulation, and the calculation formula is:

$$H = \frac{T - T_0}{g} + h \tag{16}$$

In the equation: H is the geothermal water circulation depth, m; T is the geothermal reservoir temperature, °C, taking the above calculation result; $T_0$ is the average temperature of the study area—the annual average temperature within the geographical area of Wuhan city is 16.2–16.7 °C, which, for our purposes, we will accept as 16.5 °C; g is the geothermal gradient—according to the geothermal gradient of Hubei area released by China Geothermal Flow Database, and the collected temperature data of each borehole in Wuhan city, the geothermal gradient of Wuhan city is generally 19~27.5 °C/1 km, accepted as 23 °C/1 km; h is the depth of constant temperature zone, accepted as 0.03 km. The results of geothermal water circulation depth are shown in Table 7. The geothermal reservoir temperature becomes larger with the increase of circulation depth, and the circulation depth is 1.28–4.12 km.

**Table 7.** Geothermal fluid circulation depth table of calculation results.

| Id | Heat Storage Temperature (°C) | | | Circulation Depth (km) |
|---|---|---|---|---|
| | Quartz (Non-Vapor Loss) | Quartz (Maximum Vapor Loss) | Average Value | |
| DXR-1 | 53.76 | 66.89 | 60.33 | 1.94 |
| DXR-2 | 81.32 | 91.79 | 86.56 | 3.08 |
| DXR-3 | 59.85 | 72.44 | 66.15 | 2.19 |
| DXR-4 | 59.85 | 72.44 | 66.15 | 2.19 |
| DXR-5 | 38.04 | 52.46 | 45.25 | 1.28 |
| DXR-6 | 73.45 | 84.73 | 79.09 | 2.75 |
| DXR-7 | 106.67 | 114.24 | 110.46 | 4.12 |
| DXR-8 | 76.79 | 87.72 | 82.26 | 2.89 |
| DXR-9 | 73.45 | 84.73 | 79.09 | 2.75 |
| DXR-10 | 92.30 | 101.56 | 96.93 | 3.53 |
| DXR-11 | 101.92 | 110.06 | 105.99 | 3.92 |

*4.7. Conceptual Model of the Heat Reservoir in the Research Area*

All geothermal water is supplied with atmospheric precipitation infiltration, and the regional thermal background value is not high, which belongs to the geothermal system heated by normal terrestrial heat flow. There are great differences in the hydrochemical type, age, heat storage temperature, and circulation depth of geothermal water in different geothermal reservoirs. According to the differences and common points, the genetic model can be summarized as deep sulfuric acid geothermal water and shallow bicarbonate geothermal water, and the Silurian strata with heat-water insulation in the middle. The Silurian system in the province is mainly composed of shale and sandy shale, which are widely distributed, with a thickness of 1125–3416 m. The Silurian system has poor permeability and low thermal conductivity and is a good regional water insulation layer. It can not only inhibit the depth of groundwater migration in overlying strata but also is not conducive to the upward transfer of deep heat flow, which is unfavorable to the activity conditions of water heat exchange. It acts as a thermal insulation barrier to the underlying heat reservoir, which is conducive to the generation and enrichment of hot water in the underlying rock mass. Therefore, there is a significant difference in the number of hot water points between the upper and lower strata with Silurian as the interface.

Due to the small number of data points, the Kriging interpolation method was used to interpolate the supply elevation, age, heat storage temperature, and cycle depth into the graph. The spatial interpolation of the four elements is mainly to reflect the characteristics of each geothermal well. The difference in recharge elevation and cycle depth of each sampling point can be illustrated graphically, which is conducive to classification observation. According to the analysis of Figure 7, the water age, heat storage temperature, and circulation depth of the five places located in DXR7-11 are highly similar, indicating that these five places belong to the same aquifer. The replenishment elevation is the same as that of DXR-1 in the northern mountainous area. It is inferred from the replenishment elevation that both of them are recharged by the migration to the deep layer after the precipitation in the northwestern mountainous area. The remaining DXR1-6 supply elevation is shallow, the circulation depth is shallow, the age is young, and the heat storage temperature is low. It should belong to shallow geothermal water, which is directly supplied by atmospheric precipitation. It can be preliminarily proven that the geothermal in Wuhan is mainly divided into two deep and shallow geothermal aquifers.

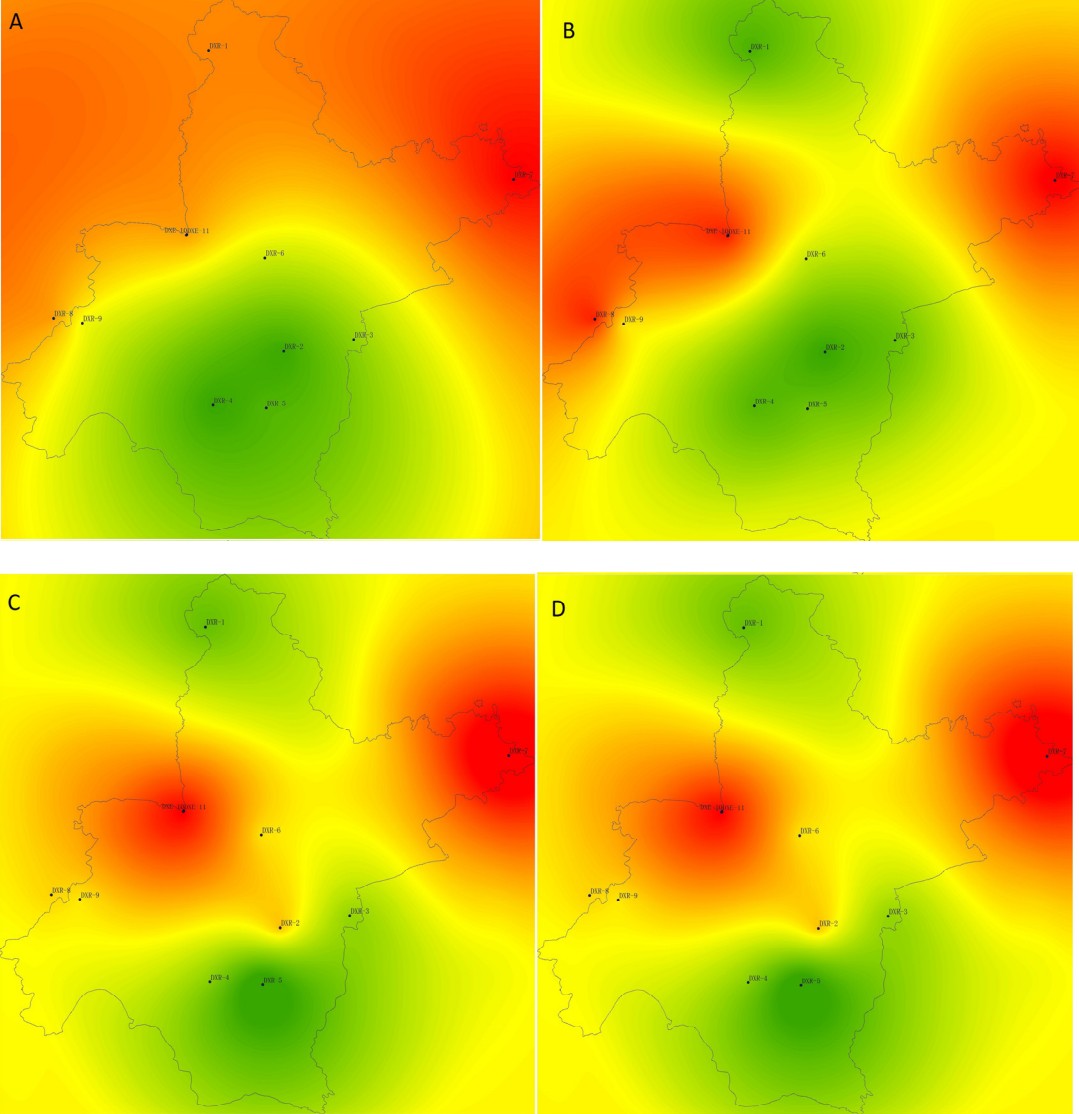

**Figure 7.** Kriging interpolation map of outcrop hot spots in Wuhan. (**A**) Recharge Elevation. (**B**) Age. (**C**) Heat storage temperature. (**D**) Depth of circulation. The value increases as the color deepens.

The first type is deep cycle high TDS sulfuric acid water: the age of geothermal water is 17~38 ka, and its circulation depth is 2.75~4.12 km. Deep geothermal water, which mainly communicates the surface and deep heat sources through faults, exists in the Sinian-Ordovician strata. Such strata are mainly found in the bedrock mountains in the north and south of Wuhan City. In the mountainous area, the lateral runoff is supplied by atmospheric precipitation, surface water, and adjacent aquifers, and it enters the deep underground through the fault structure cycle. Driven by high temperature and high pressure, the deep hot fluid migrates upward through the fractures in the cap layer. In the ascending process, the water mixed with the upper fracture zone reached a new equilibrium. After being heated by an underground heat source, it emerges from the mountain basin area of Wuhan City.

The second type of shallow circulation is low TDS bicarbonate water: the geothermal water's age is 2.3~8.9 ka, the overall circulation depth is shallow, and the geothermal water temperature is low. It is distributed in Devonian-middle lower Triassic carbonate rocks and upper Triassic-Tertiary clastic rocks. The Silurian is the base below, and the Tertiary and Quaternary sedimentary layers are attached above. There is good water communication with the surface. It is mainly fractured pore water of clastic rock, and the replenishment discharge depends on the development degree of the crack opening. The circulation pathway in Hubei Province is mainly from the northwestern Yingshan to the Wuhan basin. The main source of recharge is atmospheric precipitation, and the water flow occurs and migrates near the surface. Along with the surface pores, the water channel decreases continuously. The hot fluid in the shallow heat reservoir is driven by high-temperature pressure, and the water source is heated by the heat source and continues to migrate upward through the fracture. Due to the decrease in temperature and pressure in some sections, the precipitation of $SiO_2$ and other chemical substances is self-sealing, forming a shallow heat reservoir cap. In other areas, the hot fluid moves upward, mixing with groundwater to varying degrees on the way, which is diluted and cooled, flowing out of the surface and becoming surface geothermal. Some of the atmospheric precipitation penetrates deep through the fault. After being heated deep underground, the fault rises and gathers in the shallow layer to become a hot fluid.

According to the above inference, the general geothermal circulation model of Wuhan is shown in Figure 8. Cambrian and Ordovician strata are mainly found in Dahongshan in the northwest of Wuhan and Shogunshan in the southeast. The atmospheric precipitation in the mountainous area converges with the joint fissure in the fault zone and migrates to the depth along the formation slope in the horizontal direction. The groundwater migrates from the mountain area to the deep basin and is heated by geothermal heating and deep heat source. Under the action of pressure and temperature, the ascending fault zone is exposed to a hot spring, and part of it is supplied to a shallow geothermal reservoir. The Wanglun geothermal, Tangchi geothermal, DXR-8, and DXR-9 geothermal near Wuhan are in a trend line, and their hydrochemical characteristics are similar to each other. It is inferred that Wuhan belongs to basin conduction type geothermal. The deep geothermal is recharged by the groundwater in the marginal mountain area and migrated for a long distance. After a long heating process, it emerges into a spring in the basin. The shallow geothermal is directly supplied by local atmospheric precipitation, river infiltration, and other water sources. The Paleogene and Quaternary cap strata are poorly insulated and adequately replenished by the surface. Part of the groundwater is subjected to fracture conduction heat, while part of the groundwater is mixed with deep geothermal water and, finally, forms shallow geothermal warm water.

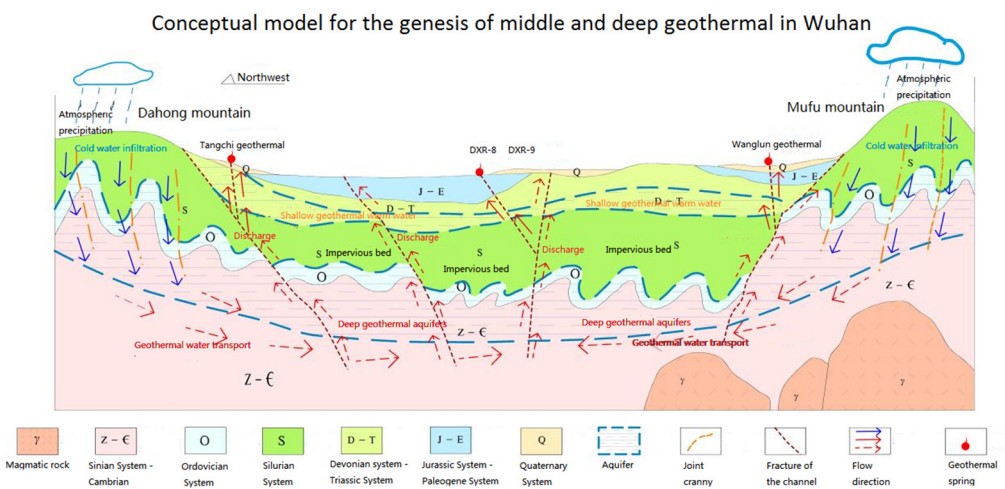

**Figure 8.** Conceptual model diagram of deep and shallow geothermal genesis in Wuhan.

## 5. Conclusions

This study mainly analyzed the chemical characteristics of Wuhan geothermal water, including the water's chemical characteristics, isotopes, heat storage temperature, and circulation depth, and proposed a conceptual model of Wuhan geothermal. Through the collected water sample information, the chemical characteristics, principal components, and correlation analysis of Wuhan geothermal water are completed, and the final comprehensive understanding of the chemical characteristics of the water of the Wuhan area. By means of isotope analysis, the recharge elevation, atmospheric precipitation recharge, and water age of Wuhan geothermal water were preliminarily analyzed. In order to identify geothermal storage temperature in the Wuhan area, the water-rock equilibrium state analysis has been preliminarily completed. Using Multiple Mineral Equilibrium Approaches, the $SiO_2$ Geothermometer Method, and the Silicon-Enthalpy Mixing Model, a comparative analysis was carried out. Among them, Multiple Mineral Equilibrium Approaches are more suitable for qualitative judgment, and the latter two approaches are closer to the actual measured value. The judgment of circulation depth also basically indicates the basic characteristics of the two main aquifers in Wuhan, but there are few water sample points, so more data are needed to further support the study and judgment. According to the chemical properties of water, it can be roughly divided into two types: shallow-cycle low-TDS bicarbonate water and deep-cycle high-TDS sulfuric acid water. Combined with the topographic structure and hydrochemical analysis results, the circulation mechanism of deep and shallow geothermal in the Wuhan area was preliminarily discussed. The conceptual genesis model of geothermal in the Wuhan area was proposed, and the distribution and chemical characteristics of geothermal water in the Wuhan area were defined.

The first category is represented by DXR-2, DXR-3, and DXR-4, with $HCO_3^-$ as the main anion and $Ca^{2+}$ as the main cation. It is presumed to be greatly influenced by the dissolution of dolomite and calcite. Groundwater receives recharge from atmospheric precipitation, with low recharge elevation, shallow circulation depth, low geothermal reservoir temperature, little influence from fractures, relatively closed stratum, relatively poor hydrodynamic conditions, low overall TDS, and high influence from shallow cold water mixing in the ascending process.

The second category is represented by DXR-8, DXR-9, DXR-10, DXR-11, and DXR-7, with anions mainly of $SO_4^{2-}$, receiving atmospheric precipitation recharge, large circulation depth, large geothermal water age, relatively high geothermal reservoir temperature after a long period of deep circulation, good stratigraphic openness, good hydrodynamic conditions, especially DXR-8 and DXR-9 are the best, fast water body alternation, more adequate water-rock interaction, and high overall TDS.

The geothermal level in Wuhan belongs to the conduction type of basin and is supplied by the marginal mountains. Through the infiltration of fractures and fissures, it migrates to the deep geothermal spot. After deep geothermal heating or heat source heating, it migrates horizontally along the strata, rises along the faults, and finally converges in the basin. Part of it is mixed with shallow groundwater to form shallow warm water with low TDS, and part of it is exposed to the surface by drilling, forming geothermal wells.

The characteristic coefficient analysis shows that the DXR-8 and DXR-9 reservoirs have high geothermal reservoir temperature, complete borehole exposure, good hydrodynamic conditions, fast water alternating degree, high TDS content, and deep circulation, which belongs to the favorable position of geothermal resources exploitation. The sampling points collected and collected in this study are limited and dispersed, so relevant work will be supplemented in future work to further study the chemical characteristics and genetic mechanism of the water of Wuhan so as to master more detailed data on Wuhan geothermals and provide a reference value for geothermal research in similar areas.

**Author Contributions:** Conceptualization, Z.Y., C.H. and X.L. (Xuan Li); methodology, Z.Y. and X.L. (Xuan Li); software, B.H., J.Z., X.L. (Xiaozhe Li), and W.H.; validation, W.C., P.H., J.Z. and C.R.; formal analysis, J.H., B.H. and W.C.; investigation, Z.Z., S.H. and L.X.; resources, C.H. and Z.Z.; data curation, Z.Y.; writing—original draft preparation, Z.Y. and X.L. (Xuan Li); writing—review and editing, Z.Y., C.H. and X.L. (Xuan Li); visualization, Z.Y. and X.L. (Xuan Li); supervision, C.H.; project administration, C.H.; funding acquisition, C.H. All authors have read and agreed to the published version of the manuscript.

**Funding:** This research was funded by the Project of Jiangxi Geological Exploration Fund Project (No. 20160007 and 20190006), Wuhan Multi factor Urban Geological Survey demonstration Project (No. WHDYS-2021-005), China Geological Survey (No. DD20211391) and Foundation of Nanchang Key Laboratory of Hydrogeology and High Quality Groundwater Resources Development and Utilization (No. 20232B21).

**Data Availability Statement:** The data presented in this study are available on request from the first or corresponding authors.

**Acknowledgments:** The Corresponding author of this manuscript is Xuan Li, and the contributions of the authors are confirmed in the research. I would like to declare on behalf of my co-authors that the work described was original research that has not been published previously and is not under consideration for publication elsewhere, in whole or in part. All the authors listed have approved the manuscript that is enclosed.

**Conflicts of Interest:** The authors declare that they have no conflict of interest. The contents of this manuscript will NOT be copyrighted, submitted, or published elsewhere while acceptance by the Journal is under consideration. There are NO directly related manuscripts or abstracts, published or unpublished, by any authors of this paper.

## Appendix A

**Table A1.** Field indicators and water chemical composition units of geothermal water samples (mg/L).

| Id | Category | Temperature °C | Elevation | Ph | TDS | $K^+$ | $Na^+$ | $Ca^{2+}$ | $Mg^{2+}$ | $HCO_3^-$ | $SO_4^{2-}$ | $Cl^-$ | $F^-$ | Soluble $SiO_2$ |
|---|---|---|---|---|---|---|---|---|---|---|---|---|---|---|
| DXR-1 | | 26 | 108 | 8.24 | 178.7 | 0.83 | 4.39 | 40.34 | 22.49 | 218.27 | 5.86 | 2.01 | 0.16 | 15.34 |
| DXR-2 | | 23 | 20.88 | 7.27 | 280.5 | 2.34 | 22.7 | 79.38 | 10.26 | 250.51 | 50.64 | 31.55 | 0.53 | 31.42 |
| DXR-3 | | 23.1 | 10.34 | 7.9 | 219.7 | 1.44 | 4.44 | 83.28 | 8.68 | 287.72 | 13.73 | 1.37 | 0.43 | 18.16 |
| DXR-4 | | 25.6 | 19.32 | 7.87 | 257.8 | 0.99 | 3.42 | 105.41 | 7.1 | 339.81 | 13.05 | 3.71 | 0.2 | 18.16 |
| DXR-5 | Geothermal | 26 | 45 | 7.42 | 315.24 | 1.34 | 14.64 | 83.37 | 14.64 | 351.65 | 12.41 | 3.17 | 0.27 | 9.63 |
| DXR-6 | water | 22.6 | 23 | 7.25 | 643.15 | 15.69 | 65.81 | 135.06 | 24.79 | 538.76 | 71.09 | 61.33 | 3.41 | 25.9 |
| DXR-7 | samples | 32 | 62.39 | 8.15 | 659.4 | 5.54 | 211.35 | 31.23 | 3.55 | 252.99 | 330.1 | 8.42 | 4.97 | 55.42 |
| DXR-8 | | 51.3 | 21 | 7.4 | 2845.07 | 13.41 | 44.84 | 545.09 | 159.47 | 162.3 | 1964.84 | 4.23 | 3.13 | 28.14 |
| DXR-9 | | 51 | 21.64 | 7.04 | 2867.57 | 12.42 | 49.9 | 542.15 | 153.85 | 153.9 | 1991.73 | 6.31 | 3.41 | 25.9 |
| DXR-10 | | 31.3 | 7.54 | 7.36 | 1553.5 | 15.51 | 219.15 | 238.79 | 70.63 | 352.21 | 758.9 | 163.9 | 2.36 | 40.56 |
| DXR-11 | | 22.9 | 5.38 | 7.43 | 1552.4 | 13.43 | 219.85 | 256.36 | 61.55 | 344.77 | 772.2 | 165.4 | 2.48 | 50.12 |
| DB-1 | | 20.3 | 17 | 7.42 | 235.42 | 3.18 | 19.03 | 46.49 | 11.67 | 154.92 | 36.03 | 21.01 | 0.21 | 10.56 |
| DB-2 | | 23.8 | 25 | 7.77 | 200.23 | 2.16 | 10.59 | 44.08 | 9.24 | 140.17 | 34.63 | 11.59 | 0.17 | 10.7 |
| DB-3 | Surface water | 22.7 | 19 | 7.85 | 194 | 6.58 | 19.28 | 33.67 | 10.21 | 127.87 | 29.15 | 22.38 | 0.53 | 7.16 |
| DB-4 | sample | 22.9 | 17 | 7.98 | 201.88 | 1.98 | 9.49 | 45.69 | 9.72 | 153.45 | 32.68 | 8.89 | 0.19 | 11.22 |
| DB-5 | | 22.9 | 16 | 7.92 | 189.88 | 1.96 | 9.13 | 43.29 | 7.78 | 145.09 | 31.36 | 7.89 | 0.17 | 10.63 |
| DB-6 | | 23 | 23 | 7.53 | 215.68 | 6.62 | 24.27 | 33.67 | 9.72 | 118.04 | 41.95 | 31.71 | 0.38 | 6.87 |

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
