# Peer review of "Analysis of the Formation Mechanism of Medium and Low-Temperature Geothermal Water in Wuhan Based on Hydrochemical Characteristics"

_water, doi:10.3390/w15020227_

Round 1

Reviewer 1 Report

see attachment.

Author Response

Dear Editors:

In response to the editor's comments, we have made the following revisions to the manuscript.

  1. In the abstract part, the geological information of the region is supplemented, and the main geological conditions of the study are briefly explained.
  2. In the introduction, some insignificant words are deleted, and the main research content and scientific problems solved are emphasized and supplemented.
  3. Redrew the geographical position of Figure 1, modified the information in the figure, enlarged the font, and gave written explanations for each part of the figure.Geothermal Wells are distinguished.
  4. The multi-mineral equilibrium approaches and Silicon-Enthalpy Mixing Model are added to compare and analyze the geothermalstorage temperature problem, and the temperatures of the three results are compared to obtain more effective heat storage temperature information.
  5. The mixing of cold water at some sampling points is analyzed and discussed in the section of Silicon-Enthalpy Mixing Model.
  6. Make one-to-one corresponding modifications to the nouns in the manuscript, and systematically correct the nouns and grammar in the context.

7.References were added to key conclusions and data.

8.The changes in Figure 1 are described in point 3. Figure 3 adjusts the technical terms and increases the font size.

9.The reasons for the anomalies of δ18O were not clear. Since it does not influence the article's conclusion, it will not be discussed here.

10."partially balance water" has been modified to "partially balance the water".

11.“maximum loss” has been modified to “maximum vapor loss”.

12.“the depth of normal temperature zone” has been modified to “the depth of constant temperature zone”.

13.Font size has been increased for Figure 5. In addition, the spatial interpolation of "Recharge Elevation" and “Cycle Depth” are of little significance. But the reserved significance: The difference of recharge elevation and cycle depth of each sampling point can be illustrated graphically, which is conducive to classification observation, and the explanation is added in the article.

This manuscript revision not only accepted the editorial comments, but also fine-tuned some structural problems in the manuscript. The manuscript as a whole should be more coordinated and perfect. We hope that editors and reviewers will give us more valuable comments in the future. We will take them seriously and look forward to your reply.

Thank you and best regards.

Yours sincerely,

Xuan Li

Reviewer 2 Report

The paper is about an important topic within the energy transition that the whole world needs to promote. It is rather well written but lacks important information and thus cannot be published unless a thorough review is performed. The abstract needs to be modified in order to include important information about geology of the studied area. About the introduction, what is the scientific question the authors want to answer? The geological setting needs to be strongly improved and the location map also, as it hardly readable. English needs to be edited. The PDF document (water-2060094-peer-review-v1.pdf) shows all the comments that have to be answered.

Author Response

(The authors gave the same response as above.)

Reviewer 3 Report

The paper is interesting for the readers

Author Response

(The authors gave the same response as above.)

Round 2

Reviewer 1 Report

The manuscript has been revised in response to comments, but there are several minor issues that require further revision as follows:

1. The left and right legends in Figure 1 are still not visible.

2. Due to the reviewer's mistake, the Chinese Meteoric Water Line in Fig. 3 is regarded as Global Meteoric Water Line (GMWL), in which case both lines should belong to "Local Meteoric Water Line (LMWL)", please make a reasonable label and pay attention to the correct English expression.

3. Line 324, please note the 14C subscript and proofread the entire text.

4. Line 354, it should be "Geothermal reservoir", not "Thermal reservoir."

Author Response

Dear reviewer:

Thank you for your valuable suggestions. I have made one-to-one modifications and replies to the questions you raised, as follows:

  1. Please refer to the latest uploaded manuscript. Figure 1 in the original manuscript has been changed. Now the legend in Figure 1 has been enlarged and clearly visible
  2. The Chinese Meteoric Water Line “LMWL(China)”in Figure 3 and Wuhan Meteoric Water Line “LMWL(Wuhan,China)" in Wuhan have been properly labeled and the correct English expression should be noted.
  3. Proofread the entire manuscript and revised all 14C, including line 324.
  4. According to the suggestions and the manuscript structure, the "Thermal reservoir" in Line 354 has been changed into "Geothermal reservoir", including other places.

Reviewer 2 Report

Comments on the revised version of the manuscript entitled Analysis of the formation mechanism of medium and low temperature geothermal water in Wuhan based on hydrochemical characteristics 

This second submission follows a thorough review and every issue raised should be answered or at least explained if no answer can be brought.

Abstract, L. 26 and following: How can the Silurian be a barrier between Cambrian and Ordovician and between Upper Triassic and Paleogene? 

L 97 : Those names have to be indicated on the map (Fig. 1), as well as all the others in order that the readers can figure out where they are located : Dabie Mountain in northeastern Hubei and the Makufu Mountain in southeastern Hubei, in the eastern part of the Jianghan Plain, Yangtze and Han rivers, etc.

L 116 and following : “It divides Wuhan into two types of underground aquifers, deep and shallow, which are Cambrian-Ordovician strata and two main aquifers of the Devonian-Third system. The lower part is based on the Silurian system, and the upper part is attached to the sedimentary layer of the Tertiary and Quaternary system, which has good water communication “. A geological cross section would be needed to show the compartments of the hydrologic system. Otherwise it is too hard to understand.

L121: supplemental diameter discharge : What does it mean? Explanation should be brought.

There is a figure with no numbering before Fig.1

Fig. 1 : what is the reference for elevation? What is the unit? -142m to +775m? If so, it would have to be indicated, at least in the figure caption. Fig 1 still does not have subdivisions A, B, C… despite what is indicated in the caption.

L130 : Where were the “six sets of surface water samples” collected : in springs, rivers, where else? Question already asked in the 1st review and not answered. 

L138 : “All sampling sites were taken for full analysis, and trace elements” No acidizing is mentioned (already indicated in the first review and not corrected): in that case, the lowering of temperature induces precipitation of dissolved salts, hence the analysis of the water performed afterwards is spoiled by this precipitation and the results must be discarded. 

So many other comments are to be made throughout the manuscript.

Overall, most of the comments that were made in the 1st review have not been addressed by the authors. No answers were brought or corrections made.

Author Response

Dear reviewer:

Thank you for your valuable suggestions. I have made one-to-one modifications and replies to the questions you raised, as follows:

  1. Abstract, L. 26 and following: How can the Silurian be a barrier between Cambrian and Ordovician and between Upper Triassic and Paleogene?Detailed analysis of the above problems will be added in "2. Geology and hydrological settings of the study area". The explanation of the problem is very detailed and easy for the reader to understand.
  2. L 97 : Dabie Mountain in northeastern Hubei and the Makufu Mountain in southeastern Hubei, in the eastern part of the Jianghan Plain, Yangtze and Han rivers, etc. These place names have been marked in the updated Figure 1 so that readers can find out exactly where they are.
  3. L 116 and below: Stratigraphic and hydrological system zoning can be referred to Figure In addition, detailed explanation and analysis are given in 4.7.
  4. L121: The sentence has been deleted and has no effect on the content of the manuscript.There are no other diagrams before Figure 1. Figure 1: The reference units for elevation are meters: -142 m to +775 m. Figure 1 is divided into two parts, A and B. Part A mainly points out the specific location of the research area in China. Part B indicates the elevation of the study area, faults, locations of sampling points, and other information.
  5. L130: Six sets of surface water samples and 11 sets of geothermal water have been added to explain where they were taken. The manuscript has answered the question raised by the reviewer.
  6. L138: The specific quantity of sampling and the preservation details of acidification during the sampling process are added.

All the questions for the second time have been answered above, and corresponding modifications have also been made in the manuscript.
